# Steering Protein Language Models

**Long-Kai Huang** [1]   **Rongyi Zhu** [1]   **Bing He** [1]   **Jianhua Yao** [1]

## Abstract

Protein Language Models (PLMs), pre-trained on extensive evolutionary data from natural proteins, have emerged as indispensable tools for protein design. While powerful, PLMs often struggle to produce proteins with precisely specified functionalities or properties due to inherent challenges in controlling their outputs. In this work, we investigate the potential of Activation Steering, a technique originally developed for controlling text generation in Large Language Models (LLMs), to direct PLMs toward generating protein sequences with targeted properties. We propose a simple yet effective method that employs activation editing to steer PLM outputs, and extend this approach to protein optimization through a novel editing site identification module. Through comprehensive experiments on lysozyme-like sequence generation and optimization, we demonstrate that our methods can be seamlessly integrated into both auto-encoding and autoregressive PLMs without requiring additional training. These results highlight a promising direction for precise protein engineering using foundation models. Code is available at Github[1].

## 1. Introduction

Protein Language Models (PLMs) (Madani et al., 2020; Nijkamp et al., 2023; Lin et al., 2022; Hayes et al., 2024; Lv et al., 2024) have emerged as transformative tools for understanding and designing proteins (Notin et al., 2022; Strokach & Kim, 2022; Ferruz & Höcker, 2022; Meier et al., 2021). By distilling evolutionary information from billions of protein sequences, these models encode rich biological knowledge about protein structure and function. Coupled with additional predictors, PLMs achieve state-of-the-art perfor-

mance in predicting diverse protein properties (Mardikoraem & Woldring, 2023; Notin et al., 2022; Zhang et al., 2024; Chu et al., 2024; Gordon et al., 2024). However, their ability to generate proteins with precisely specified properties remains limited, typically requiring massive sequence generation followed by resource-intensive screening.

To address this limitation, several strategies have been explored to control the generation. One straightforward method involves fine-tuning PLMs using a dataset consisting of proteins that exhibit desired characteristics, thereby converting a general-purpose model into a specialized generator (Nijkamp et al., 2023). This approach, however, demands hundreds or even thousands of high-quality data and substantial computational resources. Besides, it risks diluting the general knowledge encapsulated during the initial pre-training phase. Another strategy incorporates special keyword tags indicating functionalities or properties during the pre-training, aiming to guide the generation process akin to prompting in large language models (LLMs) for natural language processing (Madani et al., 2020; Lv et al., 2024). Yet, this approach lacks flexibility, as any control requires alignment with the tags used during the pre-training phase, limiting its adaptability to new controls. These challenges motivate the exploration of inference-time control methods that preserve model knowledge while enabling precise steering.

Inference-time intervention methods, known as activation steering or activation editing, have been introduced to guide the generated texts of LLMs toward desired behaviors (Subramani et al., 2022; Turner et al., 2023; Panickssery et al., 2023; Wang & Shu, 2023; Liu et al., 2023; Li et al., 2024; Zou et al., 2023; Cao et al., 2024; Qiu et al., 2024; Lee et al., 2024a). These methods presuppose that the models inherently possess the knowledge required to generate the desired output in the internal representations but may not always actualize this potential in its outputs. By modifying the internal activations, we can steer the model's behavior to produce the desired texts. Despite their success in LLMs, these techniques remain largely unexplored in the context of PLMs, where the unique characteristics of protein languages present additional challenges.

In this paper, we explore the potential of activation steering in PLMs, aiming to guide protein generation toward

---

[1]Tencent AI Lab. Correspondence to: Long-Kai Huang <hlongkai@gmail.com>, Jianhua Yao <>.

*Proceedings of the 42$^{nd}$ International Conference on Machine Learning*, Vancouver, Canada. PMLR 267, 2025. Copyright 2025 by the author(s).

[1]https://github.com/Long-Kai/Steering-PLMs

sequences with specific properties. We begin by confirming that PLMs indeed encode knowledge about these properties, as detailed in Section 3.1. Subsequently, we adapt the Activation Addition technique (Turner et al., 2023), originally developed for LLMs, to steer the outputs of both auto-regressive and auto-encoding PLMs. Specifically, we compute a steering vector as the mean difference in internal representations between proteins with and without the target property. During inference, we add this vector to the models activations, biasing generation toward proteins with the desired characteristics.

We further extend the proposed method to protein optimization tasks using auto-encoding PLMs, which predict beneficial mutations in protein sequences toward the target property. Without steering, standard models, guided by co-evolutionary patterns learned from natural proteins, tend to predict sequences similar to those found in nature. To direct these models towards generating novel proteins with desired properties, we first identify mutation sites that are crucial for achieving these properties and then apply activation steering in the prediction. To implement this, we propose a novel algorithm that selects mutation sites based on the dissimilarity between the token representations and the steering vector. By integrating this algorithm with activation steering, our method effectively guides protein sequence optimization toward target properties.

Our work makes three key contributions:

1) We present the first application of activation steering to PLMs, enabling property-specific protein generation without retraining;

2) We propose a novel protein optimization framework that integrates activation steering with mutation site identification in auto-encoding PLMs;

3) We provide comprehensive empirical validation on protein generation and optimization across diverse PLM architectures (ProLLaMA, ESM2, ESM3) and biological properties, including thermostability, solubility, and green fluorescent protein (GFP) brightness.

## 2. Related Works

**Protein Language Models** (PLMs) leverage transformer architectures to learn functional and structural patterns from evolutionary protein sequences. PLMs can be broadly categorized into auto-encoding (AE) and autoregressive (AR) architectures. AE-PLMs like ESM2 (Lin et al., 2022) and ESM3 (Hayes et al., 2024) use masked language modeling to capture bidirectional dependencies, which are essential for understanding protein sequences. AR-PLMs like ProGen (Madani et al., 2020; Nijkamp et al., 2023) adopt causal language modeling, enabling de novo protein generation. These models excel in diverse tasks: zero-shot mutation effect prediction (Meier et al., 2021), evolutionary trajectory modeling (Hie et al., 2022), and structure-aware design (Zheng et al., 2023). Their latent spaces encode biophysical properties, supporting state-of-the-art performance in fitness prediction (Hie et al., 2024) and atomic-level structure inference (Lin et al., 2023).

Recent advances extends PLMs to multitask settings (Pei et al., 2024; Lv et al., 2024) and controllable protein generation (Lv et al., 2024; Madani et al., 2020; Ferruz & Höcker, 2022). However, steering PLM outputs toward user-specified functional traits, such as increased stability or solubility, remains challenging. Unlike natural language, protein generation requires preserving structural viability and evolutionary plausibility, which limits the effectiveness of standard LLM control methods like prompting or fine-tuning. While PLMs have revolutionized protein engineering, their latent spaces remain underutilized for targeted activation-based interventions, highlighting the potential for adapting inference-time steering techniques from NLP to protein design.

**Activation Steering** modifies a models behavior at inference time by perturbing its internal activations, without training or changing model weights. In LLMs, methods like activation addition (ActAdd) (Turner et al., 2023) compute steering vectors as contrasts between activations of opposing prompts (e.g., truthful vs. deceptive) and inject these into hidden states to influence outputs. Recent works improve this approach: contrastive activation addition (CAA) (Panickssery et al., 2023) aggregates steering vectors from hundreds of contrast pairs (e.g., truthful vs. hallucinated responses) to reduce noise and improve robustness across diverse prompts. Other methods refine vector extraction through dataset-driven preferences (Li et al., 2024), optimal transport (Singh et al., 2024), conditional interventions (Qiu et al., 2024), or bi-level optimization (Cao et al., 2024), enabling more precise control. These techniques have been used to steer attributes such as style, safety, and truthfulness (Zou et al., 2023; Liu et al., 2023) with applications ranging from bias mitigation (Adila et al., 2024) to adversarial robustness (Wang & Shu, 2023).

While activation steering is well-studied in LLMs, its application to protein language models (PLMs) remains unexplored. Unlike LLMs, PLMs generate outputs that reflect biophysical functionalities rather than linguistic semantics, and their latent spaces are shaped by evolutionary and structural constraints. Existing LLM methods focus on abstract linguistic properties, while steering PLM generations requires grounding activation vectors in sequence-function relationships (e.g., stability, thermostability, or solubility). To bridge this gap, in this work, we adapt activation addition to PLMs by constructing steering vectors from protein

sequences with contrasting functional traits, and demonstrate that PLM generations can be reliably steered toward user-specified properties.

**Protein Optimization** aims to design sequences with improved functional properties while preserving structural viability. Traditional approaches, such as directed evolution (DE), rely on iterative mutation and screening (Romero & Arnold, 2009). Machine learning-assisted DE (MLDE) accelerates this process by predicting fitness from sequence data (Wu et al., 2019; Wittmann et al., 2021), but its dependence on experimental labels limits scalability (Yang et al., 2025). Zero-shot predictors, such as PLM likelihoods, partially mitigate this by estimating fitness without labeled data (Meier et al., 2021; Notin et al., 2024), but they struggle to explicitly guide generation toward desired properties.

Recent advances leverage generative PLMs for de novo protein design (Madani et al., 2023; Nijkamp et al., 2023) and latent space optimization (Stanton et al., 2022; Kirjner et al., 2023). For example, (Kirjner et al., 2023) proposed to smooth noisy fitness landscapes using energy-based models to guide Gibbs-with-Gradients sampling, while (Stanton et al., 2022) proposed to optimize sequences in latent space via denoising autoencoders. However, these methods require training differentiable fitness proxies or imposing explicit structural constraints, limiting their flexibility. As for reinforcement learning (Lee et al., 2024b; Angermueller et al., 2019) and evolutionary algorithms (Ren et al., 2022) based methods, they both face trade-offs between exploration and computational cost.

In contrast, activation steering offers a lightweight alternative. By perturbing PLM activations at inference time, we bypass weight updates and expensive sampling and directly inject functional preferences into generation. Unlike previous PLM-based optimization methods, which use likelihoods as proxies or fine-tune models on labeled data, our method directly steers generation toward target properties without explicit fitness predictors.

# 3. Activation Steering for PLMs

## 3.1. Premise Verification

Protein language models (PLMs) have been shown to provide effective representations for downstream property prediction tasks (Notin et al., 2022), indicating that they capture relevant functional and structural information. This capability forms the foundation for activation steering, where model activations are manipulated to influence the property of the generated proteins.

To support this premise, we analyze the internal representations of several PLMs. As shown in Figure 1, t-SNE visualizations of activations from ESM2, ESM3, and ProLLaMA

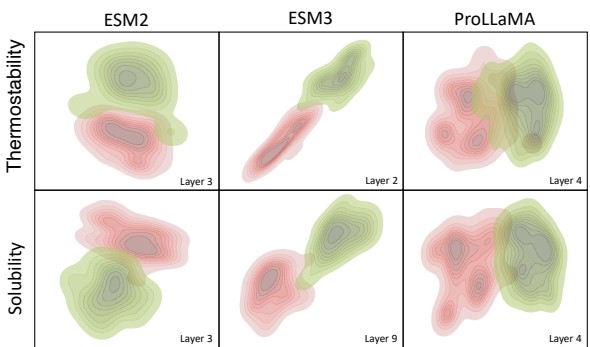

*Figure 1.* t-SNE visualization of PLM activations from ESM2, ESM3, and ProLLaMA for proteins with (red) and without (green) the target properties: high thermostability (top row) or high solubility (bottom row). Partial separation of clusters suggests that property-related information is encoded in the activation space.

reveals that proteins with and without target properties form partially distinct clusters in activation space. This pattern is consistent across different models. These observations confirm that PLMs inherently encapsulate intrinsic knowledge about specific protein properties and motivate our approach to steering protein generation toward desired characteristics through activation space manipulation.

## 3.2. Activation Steering for PLMs

To guide protein generation toward desired properties, we employ activation steering, which modifies model activations using steering vectors. Specifically, at each layer $l$, the activation is edited as:

$$\tilde{\mathbf{h}}_l = \mathbf{h}_l + \alpha \mathbf{v}_l, \tag{1}$$

where $\mathbf{h}_l$ and $\tilde{\mathbf{h}}_l$ are the activations in the $l$-th layer before and after steering, respectively, $\mathbf{v}_l$ is the steering vector for the $l$-th layer, and $\alpha$ is a scalar hyper-parameter controlling the steering strength. After modification, the edited activations $\tilde{\mathbf{h}}_l$ is rescaled to have the same norm as $\mathbf{h}_l$ before being passed to the next layer. The steering process is illustrated in Figure 2b. We apply activation steering to all layers except the input layer and across all tokens.

The steering vectors used in Equation (1) are computed as the mean difference in representations at the $l$-th layer between sets of proteins with and without the desired property. These steering vectors point in the direction from undesired to desired properties. For AE-PLMs, we use the average activations across all tokens; for AR-PLMs, we use the last token's activation. Formally, for AE-PLMs, the steering vectors are calculated as:

$$\mathbf{v}_l = \frac{1}{|\mathcal{P}|} \sum_{x_p \in \mathcal{P}} \mathbf{h}_l^{avg}(x_p) - \frac{1}{|\mathcal{N}|} \sum_{x_n \in \mathcal{N}} \mathbf{h}_l^{avg}(x_n), \tag{2}$$

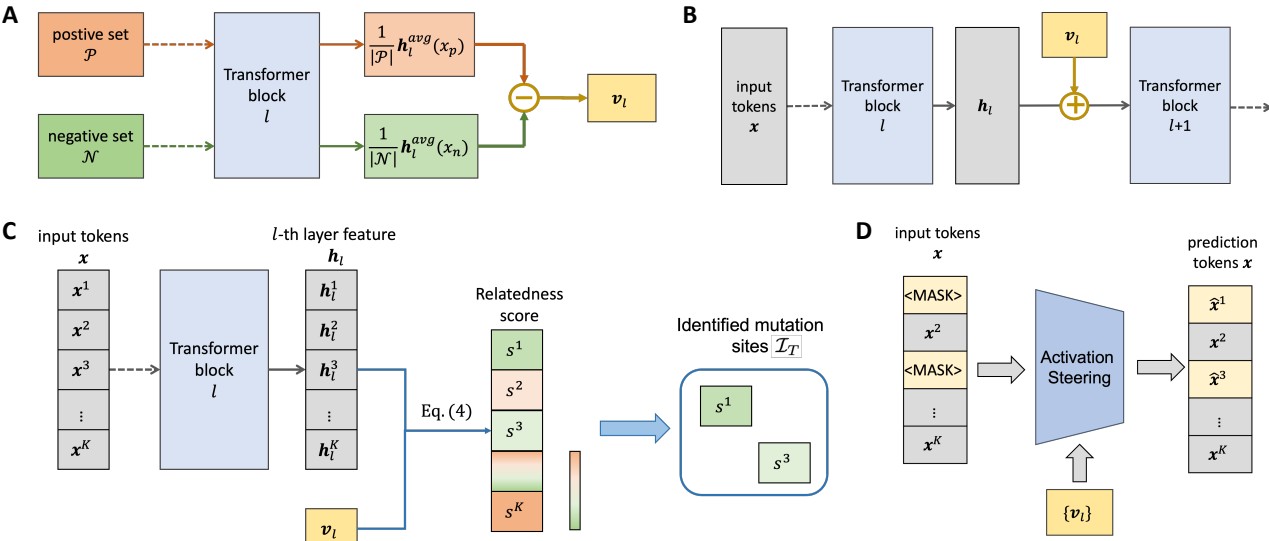

*Figure 2.* Overview of Activation Steering for PLMs and Activation Steering based Protein Optimization (ASPO). (a) **Computation of steering vectors**: For each layer, the steering vector is computed as the mean difference in activations between positive (desired property) and negative (undesired property) protein sets. (b) Activation steering during generation: At each layer, model activations are modified by adding a scaled steering vector. (c) **Identification of mutation sites**: For a given protein, token representations at a selected layer are projected onto the steering vector to compute relatedness scores. Tokens with the lowest scores (most negatively related to the target property) are selected as candidate mutation sites. (d) **Mutation prediction**: Identified mutation sites are masked, and new amino acids are predicted using activation steering.

and for AR-PLMs:

$$\mathbf{v}_l = \frac{1}{|\mathcal{P}|} \sum_{x_p \in \mathcal{P}} \mathbf{h}_l^{last}(x_p) - \frac{1}{|\mathcal{N}|} \sum_{x_n \in \mathcal{N}} \mathbf{h}_l^{last}(x_n), \quad (3)$$

where $\mathcal{P}$ and $\mathcal{N}$ are the positive and negative sets of proteins regarding the desired property, respectively. They are the sets of proteins with and without the desired property. $\mathbf{h}_l^{avg}(x_p)$ and $\mathbf{h}_l^{last}(x_p)$ denote the average activation of all tokens and last token activation of the $l$-th layer for a sequence input $x$, respectively. The computation of steering vectors is illustrated in Figure 2a.

### 3.3. Protein Optimization via Activation Steering

AE-PLMs predict the amino acid (AA) at a masked position in a protein sequence using the contextual information of surrounding AAs. While this mechanism leverages coevolutionary patterns from natural proteins, it does not directly optimize for specific target properties. To optimize a protein for desired properties, we propose to identify and mutate tokens that are negatively related to the target property, and apply activation steering to guide PLM's prediction at these positions toward desired properties.

A key challenge is to systematically identify which AAs in a sequence are most opposed to the target property. We address this by leveraging the steering vector $\mathbf{v}_l$, which en-

codes the direction of the desired property in the models representation space. For each token, we compute a *relatedness score* as the cosine similarity of its representation $\mathbf{h}_l^k$ to $\mathbf{v}_l$:

$$s^k = \text{sim}(\mathbf{h}_l^k, \mathbf{v}_l) = \frac{\mathbf{v}_l^\top \mathbf{h}_l^k}{\|\mathbf{v}_l\|\|\mathbf{h}_l^k\|}, \quad (4)$$

where $s^k$ is the relatedness score of the $k$-th token in the $l$-th layer. The projection of a tokens representation onto $\mathbf{v}_l$ quantifies relatedness to the target property. Tokens with large positive projection (large $s^k$) indicate their corresponding AAs are strongly related to the property, while those with large negative projections (small $s^k$) are less related or even opposed. After computing the relatedness scores for all tokens, we rank these scores and select the $T$ tokens with the lowest values (i.e., most negatively related to the target property) as mutation sites. The PLM then predicts new amino acids for these positions, guided by activation steering. This process is illustrated in Figure 2c and Figure 2d.

To compute the relatedness score, we need to identify the most informative layer. Specifically, we split the positive ($\mathcal{P}$) and negative ($\mathcal{N}$) sets into training and validation subsets. For each layer, we train a linear classifier on the training subset to distinguish between representations from $\mathcal{P}$ and $\mathcal{N}$, and evaluate the validation accuracy on the validation subsets. We then compute the relatedness scores for the

---

**Algorithm 1** **A**ctivation **S**teering based **P**rotein **O**ptimization (**ASPO**)

---

1: **Input:** protein sequence $x$, positive protein sequence set $\mathcal{P}$, negative set $\mathcal{N}$, steering strength $\alpha$, layer $\ell$ for relatedness score computation, number of mutation sites per round $T$, and number of rounds $R$
2: Compute steering vectors $\{\mathbf{v}_l\}$ for all layers $l = 1, 2, ..., L$ using Equation (3)
3: **for** $r = 1$ **to** $R$ **do**
4:     Compute token representations $\mathbf{h}_\ell^k$ for all tokens $k = 1, 2, ..., K$ at layer $\ell$.
5:     Compute the relatedness scores $s^k$ for all tokens using Equation (4).
6:     Obtain the set of the token indices of the $T$ lowest scores in $\{s_\ell^k\}$ as $\mathcal{I}_T$.
7:     Mask tokens at positions in $\mathcal{I}_T$.
8:     Predict new amino acids at positions in $\mathcal{I}_T$ using activation steering (Equation (1)) with steering vectors $\{\mathbf{v}_l\}$.
9: **end for**

---

layer with the highest validation accuracy and use them to select the mutation sites.

The mutation process is repeated for $R$ rounds to progressively steer the protein towards the desired properties. In each round, we compute token representations, calculate their relatedness scores, and select the mutation sites. We then mask the tokens in the selected positions and predict new amino acids using activation steering. We refer to this method as **A**ctivation **S**teering based **P**rotein **O**ptimization (**ASPO**), summarized in Algorithm 1.

## 4. Experiments

In this section, we evaluate the effectiveness of our activation steering method to control protein language models (PLMs) for property-driven protein generation and optimization.

### 4.1. Steering PLMs for Protein Generation

#### 4.1.1. EXPERIMENTAL SETTINGS

**Tasks:** We focus on the generation of lysozyme-like proteins with enhanced thermostability or solubility by steering PLMs toward these properties.

**Base Models:** We assess the effectiveness of our method across two types of PLMs: 1) auto-encoding PLMs (AE-PLMs), including ESM2 (650M) (Lin et al., 2022) and ESM3-open (1.4B) (Hayes et al., 2024); and 2) auto-regressive PLMs (AR-PLMs), including ProLLaMA (7B) (Lv et al., 2024). ProLLaMA enables controlled sequence generation via superfamily descriptions, which we use to restrict outputs to the lysozyme-like family. For AE-PLMs, which do not generate sequences directly, we start from a reference sequence and, in each iteration, randomly select and mask 10% of the tokens without replacement, then regenerate them using the model.

**Evaluation Metrics:** To evaluate all methods, we assess their generated sequences for target property fitness, diversity, and novelty. For each metric, we calculate the value

for each protein sequence and report the average with standard deviation. Detailed metric definitions are provided in Appendix A.1.

**Data:** To construct the positive and negative sets for steering vector extraction, we first predict thermostability or solubility for all lysozyme-like proteins in the UniRef50 dataset using property-specific predictors. For thermostability, sequences with predicted values above 70°C form the high thermostability set, and those below 50°C form the low thermostability set. For solubility, we construct a high solubility set using the sequences with predicted soluble probability higher than $0.8$ and a low solubility set using the sequences with predicted soluble probability lower than $0.15$. We randomly sample sequences from each high and low set to form the positive ($\mathcal{P}$) and negative ($\mathcal{N}$) sets, respectively.

**Hyper-parameter settings:** We fix positive and negative set sizes for steering vector extraction at $100$ and set $\alpha = 1.0$ by default. The sensitivity of these hyperparameters will be explored in Section 4.3.

**Baselines:** We compare our method to two baselines: (1) PLMs fine-tuned on positive sets (Fine-tuning), and (2) the original, unmodified models (Original Model). For AE-PLMs, we fine-tune only the last layer. For AR-PLMs, we use LoRA (Hu et al., 2022) on all layers with rank 4 and alpha 16. To evaluate performance, we generate 1000 sequences for each method. For AE-PLMs, we randomly select 1000 lysozyme-like proteins from the UniRef50 dataset as initial reference sequences to generate 1000 sequences.

#### 4.1.2. RESULTS AND ANALYSIS

Table 1 summarizes the performance of activation steering compared to fine-tuning and the original models across both auto-regressive (ProLLaMA) and auto-encoding (ESM2, ESM3) PLMs. For both thermostability and solubility tasks, activation steering consistently outperforms the baselines in terms of the target property. For example, on ESM2, activation steering achieves a thermostability of 82.20 compared to 63.56 for fine-tuning and 56.48 for the original model. Similar improvements are observed for solubility, where ac-

*Table 1.* Comparison of generating lysozyme-like protein with high thermostability or solubility. Results are reported as mean (std) for each metric.

| Base Model | Method | Thermostability | | | Solubility | | |
|---|---|---|---|---|---|---|---|
| | | Thermostability ↑ | Diversity ↑ | Novelty ↑ | Solubility ↑ | Diversity ↑ | Novelty ↑ |
| ProLlama | Original Model | 56.18 (8.05) | 0.931 (0.035) | 0.767 (0.064) | 0.230 (0.085) | 0.931 (0.035) | 0.767 (0.064) |
| | Fine-tuning | 57.24 (8.64) | **0.958** (0.017) | 0.798 (0.068) | 0.241 (0.086) | 0.958 (0.017) | 0.838 (0.059) |
| | Activation Steering | **67.68 (12.86)** | 0.927 (0.027) | **0.807 (0.063)** | **0.276 (0.110)** | **0.964 (0.016)** | **0.882 (0.056)** |
| ESM2 | Original Model | 56.48 (12.04) | 0.954 (0.023) | 0.591 (0.110) | 0.289 (0.151) | 0.963 (0.019) | 0.596 (0.130) |
| | Fine-tuning | 63.56 (14.87) | 0.953 (0.023) | 0.585 (0.099) | 0.356 (0.213) | 0.961 (0.020) | 0.594 (0.132) |
| | Activation Steering | **82.20 (12.92)** | **0.971 (0.023)** | **0.739 (0.130)** | **0.494 (0.241)** | **0.998 (0.001)** | **0.880 (0.087)** |
| ESM3 | Original Model | 55.20 (11.14) | 0.952 (0.021) | 0.573 (0.100) | 0.257 (0.177) | 0.958 (0.017) | 0.579 (0.123) |
| | Fine-tuning | 62.82 (14.72) | 0.949 (0.021) | 0.568 (0.104) | 0.318 (0.215) | 0.955 (0.017) | 0.570 (0.119) |
| | Activation Steering | **82.06 (12.06)** | **0.954 (0.019)** | **0.614 (0.115)** | **0.582 (0.264)** | **0.966 (0.019)** | **0.639 (0.123)** |

*Table 2.* Comparison of lysozyme-like protein optimization toward high thermostability. Results are reported as mean (std).

| | Medium difficulty | | | | Hard difficulty | | | |
|---|---|---|---|---|---|---|---|---|
| | Fitness ↑ | Diversity | $\text{Dissim}_{\text{init}}$ | $\text{Dissim}_{\text{high}}$ | Fitness ↑ | Diversity | $\text{Dissim}_{\text{init}}$ | $\text{Dissim}_{\text{high}}$ |
| Before Optimization | 59.78 (3.04) | 0.879 (0.072) | 0 | 0.601 (0.100) | 46.38 (3.11) | 0.923 (0.038) | 0 | 0.708 (0.087) |
| AdaLead | 63.56 (11.94) | 0.947 (0.036) | 0.351 (0.166) | 0.697 (0.096) | 55.16 (9.29) | 0.962 (0.018) | 0.626 (0.185) | 0.832 (0.069) |
| PEX | 66.80 (10.95) | 0.923 (0.053) | 0.203 (0.087) | 0.651 (0.086) | 48.95 (5.75) | 0.959 (0.022) | 0.185 (0.094) | 0.741 (0.073) |
| GWG | 68.25 (9.35) | 0.885 (0.068) | 0.059 (0.024) | 0.611 (0.096) | 47.73 (3.90) | 0.926 (0.036) | 0.049 (0.010) | 0.708 (0.081) |
| ESM2 + ASPO | **84.34** (7.59) | 0.840 (0.064) | 0.290 (0.167) | 0.661 (0.114) | **74.69** (12.32) | 0.828 (0.062) | 0.291 (0.163) | 0.734 (0.076) |
| ESM3 + ASPO | **88.42** (3.98) | 0.803 (0.060) | 0.110 (0.055) | 0.603 (0.077) | **86.43** (9.02) | 0.865 (0.048) | 0.161 (0.093) | 0.714 (0.072) |

*Table 3.* Comparison of lysozyme-like protein optimization toward high solubility. Results are reported as mean (std) for each metric.

| | Medium difficulty | | | | Hard difficulty | | | |
|---|---|---|---|---|---|---|---|---|
| | Fitness ↑ | Diversity | $\text{Dissim}_{\text{init}}$ | $\text{Dissim}_{\text{high}}$ | Fitness ↑ | Diversity | $\text{Dissim}_{\text{init}}$ | $\text{Dissim}_{\text{high}}$ |
| Before Optimization | 0.278 (0.012) | 0.898 (0.054) | 0 | 0.684 (0.108) | 0.085 (0.011) | 0.896 (0.056) | 0 | 0.689 (0.097) |
| AdaLead | **0.617** (0.247) | 0.949 (0.021) | 0.475 (0.162) | 0.794 (0.079) | **0.530** (0.283) | 0.959 (0.017) | 0.512 (0.177) | 0.792 (0.080) |
| PEX | 0.489 (0.246) | 0.920 (0.044) | 0.080 (0.032) | 0.711 (0.101) | 0.252 (0.240) | 0.927 (0.041) | 0.096 (0.043) | 0.715 (0.089) |
| GWG | 0.356 (0.115) | 0.912 (0.042) | 0.060 (0.019) | 0.694 (0.101) | 0.165 (0.130) | 0.919 (0.041) | 0.071 (0.030) | 0.707 (0.089) |
| ESM2 + ASPO | 0.510 (0.282) | 0.860 (0.061) | 0.022 (0.011) | 0.724 (0.105) | 0.349 (0.273) | 0.838 (0.052) | 0.018 (0.011) | 0.710 (0.063) |
| ESM3 + ASPO | **0.654** (0.273) | 0.879 (0.054) | 0.058 (0.039) | 0.720 (0.109) | **0.397** (0.247) | 0.858 (0.055) | 0.057 (0.038) | 0.711 (0.060) |

*Table 4.* Comparison of GFP optimization toward high fluorescence brightness. Results are reported as mean (std) for each metric.

| | Medium difficulty | | | | Hard difficulty | | | |
|---|---|---|---|---|---|---|---|---|
| | Fitness ↑ | Diversity | $\text{Dissim}_{\text{init}}$ | $\text{Dissim}_{\text{high}}$ | Fitness ↑ | Diversity | $\text{Dissim}_{\text{init}}$ | $\text{Dissim}_{\text{high}}$ |
| Before Optimization | 1.494 (0.340) | 0.717 (0.002) | 0 | 0.028 (0.005) | 1.325 (0.279) | 0.560 (0.002) | 0 | 0.032 (0.005) |
| AdaLead | 1.179 (0.329) | 0.737 (0.024) | 0.060 (0.088) | 0.085 (0.085) | 1.255 (0.372) | 0.596 (0.041) | 0.071 (0.095) | 0.099 (0.092) |
| PEX | 1.426 (0.337) | 0.719 (0.002) | 0.004 (0.004) | 0.032 (0.007) | 1.320 (0.298) | 0.563 (0.003) | 0.004 (0.004) | 0.036 (0.007) |
| GWG | 1.683 (0.641) | 0.721 (0.003) | 0.021 (0.002) | 0.039 (0.010) | 1.510 (0.545) | 0.568 (0.004) | 0.021 (0.002) | 0.043 (0.009) |
| ESM2 + ASPO | **3.862** (0.329) | 0.397 (0.005) | 0.020 (0.007) | 0.010 (0.007) | **3.907** (0.247) | 0.406 (0.006) | 0.022 (0.009) | 0.012 (0.009) |
| ESM3 + ASPO | **3.739** (0.357) | 0.503 (0.004) | 0.021 (0.007) | 0.010 (0.007) | **3.687** (0.321) | 0.507 (0.005) | 0.024 (0.009) | 0.012 (0.008) |

tivation steering reaches 0.494, substantially higher than the baselines. These results demonstrate that activation steering is highly effective at guiding PLMs to generate protein sequences with desired properties.

In addition to property optimization, activation steering maintains or even improves sequence diversity and novelty. For instance, on ESM3, activation steering achieves the highest solubility (0.582) while also increasing diversity (0.966) and novelty (0.639) compared to the original and fine-tuned models. This indicates that our method does not simply memorize or overfit to the positive set, but is capable of generating a broad range of novel and diverse sequences. Overall, these results highlight the advantage of activation steering for controllable and diverse protein design.

Additional experimental results for our method using ESM-3B and steering multiple target properties are provided in Appendix B.

## 4.2. Steering AE-PLMs for Protein Optimization

### 4.2.1. EXPERIMENTAL SETTINGS

**Tasks:** We evaluate our method on three protein optimization tasks: improving thermostability, solubility, and the fluorescence intensity of Green Fluorescent Protein (GFP). For thermostability and solubility, we focus on lysozyme-like proteins. For GFP, we follow the established setup in (Ren et al., 2022; Kirjner et al., 2023).

**Evaluation Metrics:** We assess all methods using four metrics: target property fitness, diversity, dissimilarity to the initial set (Dissim$_{init}$), and dissimilarity to the high-fitness set (Dissim$_{high}$). Detailed definitions of these metrics are provided in Appendix A.1.

As noted by (Kirjner et al., 2023), higher diversity and dissimilarity to the initial set do not necessarily equate to superior performance in protein optimization. Similarly, while high Dissim$_{high}$ suggests optimization toward the high-fitness reference set, it does not guarantee discovery of all possible high-fitness proteins. Therefore, it is possible for a method that generates high-fitness proteins but achieves just a fair value of Dissim$_{high}$.

**Data:** For thermostability and solubility, we use the same positive and negative sets as in our protein generation experiments. For GFP brightness, we adopt the same data split as (Kirjner et al., 2023) and randomly select 100 sequences from easy difficulty as the positive set and 100 sequences from hard difficulty as the negative set. We follow (Kirjner et al., 2023) to construct a medium difficulty task and a hard difficulty task for each target property. Each task has an initial set for optimization. For thermostability optimization, the hard difficulty task uses sequences from the low thermostability set as the initial set, while the medium difficulty task uses sequences with predicted thermostability between 50°C and 70°C. For solubility optimization, the hard difficulty task uses sequences from the low solubility set, and the medium difficulty task uses sequences with predicted soluble probability between 0.3 and 0.6. To estimate Dissim$_{high}$, we use the high thermostability set and high solubility set as the reference high-fitness set. For GFP brightness, both initial and high-fitness sets follow (Kirjner et al., 2023).

**Baselines:** We compare our Activation Steering-based Protein Optimization (ASPO) method, implemented on ESM2 and ESM3, against AdaLead (Sinai et al., 2020), PEX (Ren et al., 2022), and GGS (Kirjner et al., 2023). These baseline methods all require a surrogate fitness predictor. Therefore, we train the surrogate fitness predictor using the positive and negative sets for steering vector extraction. Note that the original AdaLead and PEX both update the surrogate fitness predictor using the generated sequences with ground-truth fitness obtained from wet-lab experiments in each round. To ensure a fair comparison, we assume no access to ground-truth fitness during optimization and do not update the surrogate fitness predictor for these methods.

**Hyper-parameter settings:** We use the same default settings as the experiments for protein generation to set the positive set and negative set sizes as 100 and steering strength $\alpha = 1.0$. For protein optimization specific hyper-parameters, we set the number of optimization rounds $R = 8$ and the number of mutation sites per round $T = 4$ for thermostability experiments and set $R = 4$ and $T = 2$ for the solubility and GFP brightness experiments.

### 4.2.2. RESULTS AND ANALYSIS

**Thermostability Optimization.** Table 2 shows the results for optimizing the thermostability of lysozyme-like proteins. Our ASPO methods, ESM2+ASPO and ESM3+ASPO, achieve the highest fitness scores across medium and hard difficulty settings. Specifically, ESM3+ASPO attains a fitness of 88.42 (medium) and 86.43 (hard), significantly outperforming all baselines. While AdaLead and PEX improve fitness over the initial set, their gains are notably smaller. In terms of diversity and dissimilarity metrics, ASPO methods yield slightly lower diversity compared to baselines, but achieve low dissimilarity to the initial set and maintain competitive dissimilarity to the high-fitness sets. This suggests that ASPO effectively steers sequences toward high-fitness regions without excessive exploration, focusing optimization on relevant sequence space.

**Solubility Optimization.** Table 3 presents results for solubility optimization. ASPO (especially ESM3+ASPO) achieves the highest or competitive fitness in both medium and hard solubility tasks, with fitness values of 0.654 and 0.397, respectively. AdaLead also performs well in terms of fitness, but at the cost of dissimilarity to the initial set, indicating a broader but less targeted search. In contrast, ASPO methods maintain lower dissimilarity to the initial set, reflecting a more focused optimization process. These results demonstrate that ASPO can efficiently improve solubility while generating sequences that are not excessively divergent from the starting set.

**GFP Brightness Optimization.** Table 4 presents results for GFP fluorescence brightness optimization. Here, both ESM2+ASPO and ESM3+ASPO achieve a significant increase in fitness, reaching 3.862 and 3.739 for the medium difficulty task, and 3.907 and 3.687 for the hard difficulty task, respectively. These values are more than double those of the best-performing baselines. Although the diversity of ASPO-generated sequences is lower, the marked improvement in the target property demonstrates the methods strong optimization capability. Dissimilarity metrics remain low, suggesting that ASPO finds high-fitness variants close to the initial set, which may be advantageous for practical protein engineering where minimal sequence changes are preferred.

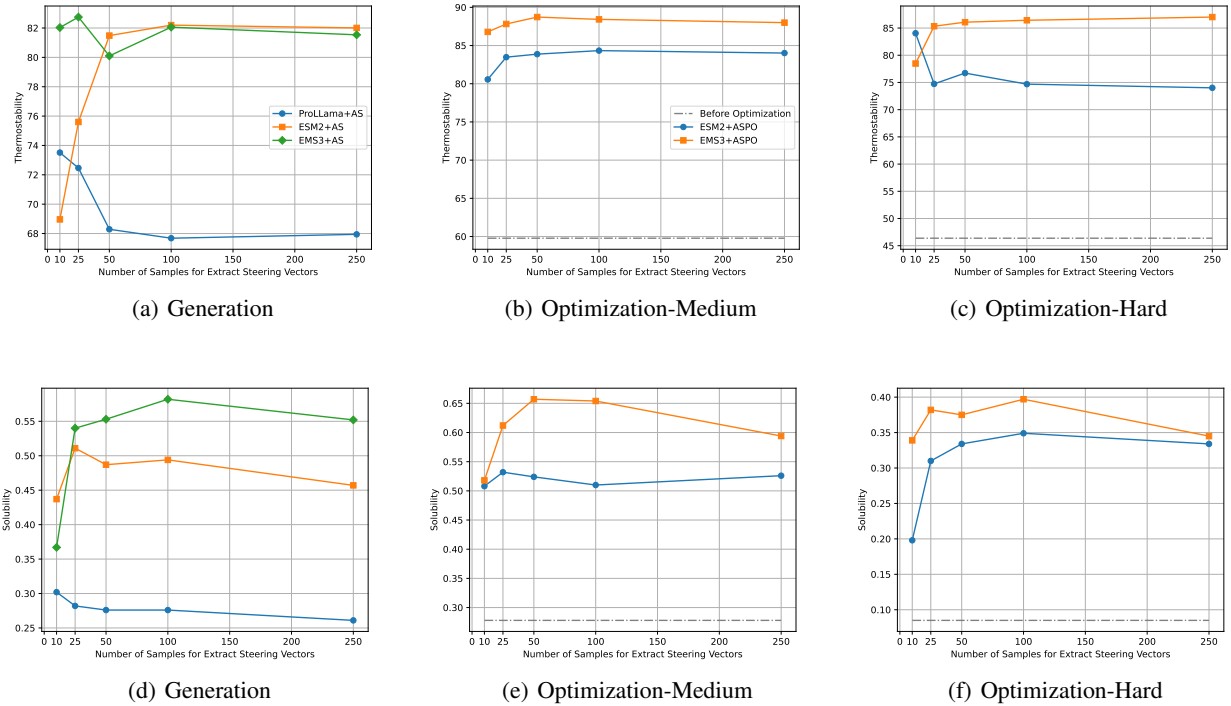

*Figure 3.* Sensitivity to the number of samples used for steering vector extraction. Columns show results for protein generation, medium-difficulty protein optimization, and hard-difficulty protein optimization. The first row is for thermostability, and the second row is for solubility.

**Summary.** Across all tasks and difficulty levels, ASPO consistently achieves the highest or near-highest fitness values. These results collectively demonstrate that integrating activation steering with mutation site identification enables more precise and reliable protein optimization compared to existing search-based methods.

### 4.3. Sensitivity to Hyperparameters

This section analyzes the sensitivity of our activation steering method to two key hyperparameters: 1) the size of the positive and negative sets used for steering vector extraction, and 2) the hyperparameter $\alpha$ used for controlling the steering strength in activation addition. We evaluate their impact on both protein generation and optimization tasks.

**Sensitivity to the positive and negative sets size.** We investigate how the size of the positive and negative sets, ranging from 10 to 250 samples, affects the effectiveness of the proposed activation steering across different model architectures and tasks.

The findings, illustrated in Figure 3, reveal varying trends and optimal set sizes depending on the architecture and task. For ESM2 and ESM3, performance generally improves as the set size increases from 10 to 100 samples, stabilizing

near peak values as the number of samples continues to grow, except for the ESM2+ASPO in the hard difficulty task for thermostability optimization. In contrast, ProLLaMA's peak performance is achieved with just 10 samples, after which there is a gradual decline. This pattern suggests that the bidirectional attention mechanisms of AE-PLMs benefit from larger, more diverse example sets to establish robust steering directions, whereas the average feature extracted from the last token in AR-PLMs may become less effective with larger sets, potentially due to increased divergence in the examples.

These observations offer practical guidelines for setting the size of the positive and negative sets. Using 100 samples generally provides reliable performance across various tasks and architectures, with diminishing returns observed beyond this point. Specifically, for activation steering on ProLLaMA in protein generation tasks, this configuration preserves over 90% of the maximum fitness performance. For ESM2 and ESM3, it captures more than 95% of the maximum potential gains. Thus, we recommend a default setting of 100 samples for these experiments.

**Sensitivity to steering strength ($\alpha$).** We investigate the impact of the steering strength $\alpha$, varying it from 0.05 to 20, across different tasks and models.

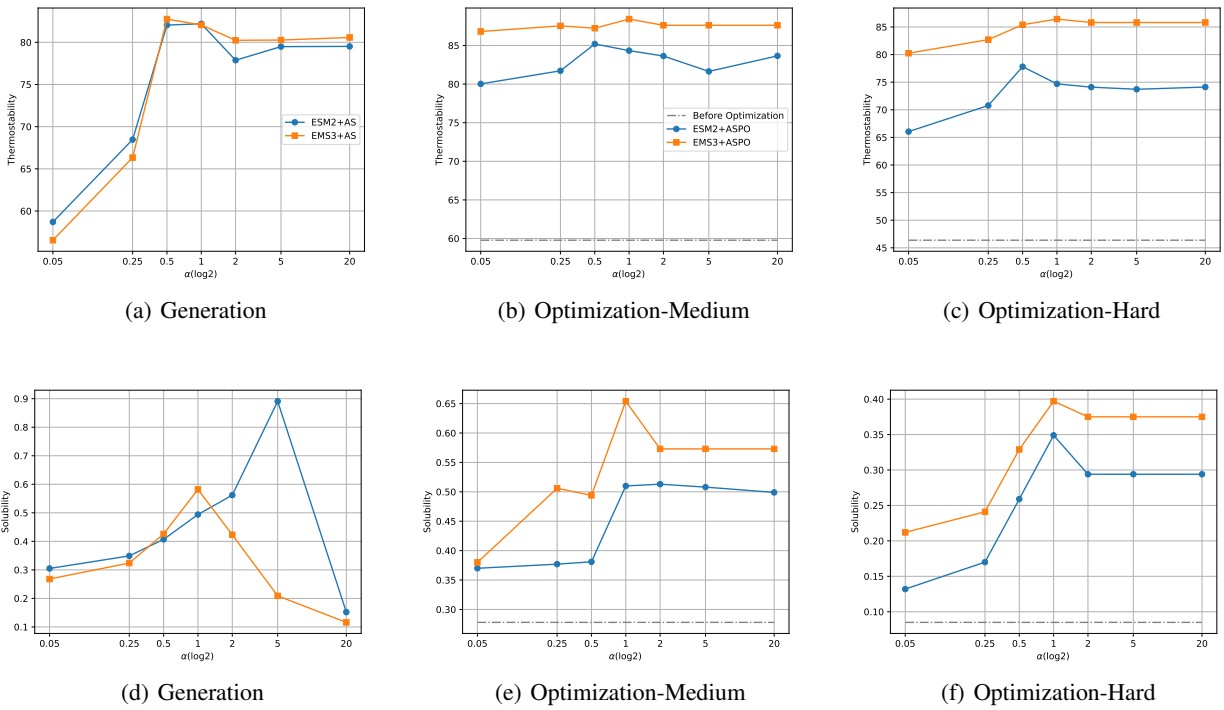

Figure 4. Sensitivity of steering strength $\alpha$. Columns show results for protein generation, medium-difficulty protein optimization, and hard-difficulty protein optimization. The first row is for thermostability, and the second row is for solubility..

As shown in Figure 4, the steering strength $\alpha$ exhibits task-dependent landscapes. For protein generation for enhanced thermostability (Fig. 4.2.2), both ESM2 and ESM3 achieves peak performance at $\alpha = 0.5$ and remain stable for $\alpha \geq 0.5$. This suggests that auto-encoding models benefit from moderate steering strength.

In the case of protein generation for enhanced solubility (Fig. 4.2.2), however, performance drops sharply for large $\alpha$ values. For both ESM2 and ESM3, we observe that their performance collapses completely at $\alpha = 20$, indicating in solubility task, the performance is more sensitive to over-steering than thermostability task. Generally over-steering ($\alpha > 5$) catastrophically degrades solubility performance while only mildly impacting thermostability.

Given these observations, we recommend a default $\alpha = 1.0$ for most applications, achieving 90-98% of maximum performance across tasks while avoiding performance cliffs. Practitioners may consider lowering to $\alpha = 0.5$ for solubility-focused applications or raising to $\alpha = 2.0$ for thermostability optimization.

## 5. Conclusion

In this paper, we demonstrate the viability of activation steering as a powerful paradigm for guiding PLM toward generating and optimizing proteins with desired properties. By deriving steering vectors from contrasting protein sets and applying them to perturb PLM activations during inference, our method enables precise, training-free control over sequence generation. Our Activation Steering-based Protein Optimization (ASPO) framework further enhances protein engineering by integrating activation editing with mutation site identification. Because our approach does not require model retraining or explicit fitness predictors, it offers a scalable and efficient alternative to traditional methods such as directed evolution or reinforcement learning based protein optimization. Ultimately, activation steering provides a promising direction for programmable protein design using foundation models.

## Acknowledgements

LK Huang would like to thank Kangfei Zhao for the support during the writing of this paper.

## Impact Statement

This paper presents work whose goal is to advance the interdisciplinary field of machine learning and protein design. There are many potential societal consequences of our work, none of which we feel must be specifically highlighted here.

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

# A. Supplementary Experimental Details

## A.1. Definition of Metrics

**Fitness** quantifies how well a protein exhibits the desired properties. We estimate fitness using predictors described in Appendix A.2 for thermostability and solubility. For GFP, we use the predictor from (Kirjner et al., 2023) to estimate the log fluorescence intensity.

**Dissimilarity Score** measures how different two sequences are. In our experiment, we estimate the dissimilarity in a set-wise manner using MMseqs2 (Steinegger & Söding, 2017). Specifically, given a query set and a target set, we use MMseqs2 to align sequences from the query set to the target set, with a maximum E-value of $10.0$ ($-e$ $10.0$) and a sensitivity of $15.0$ ($-s$ $15.0$). For each matched pair, we define dissimilarity as $1.0 -$ percent identity, where percent identity is reported by MMseqs2. For pairs with no match (not reported by MMseqs2), we assign a dissimilarity of $1.0$.

**Diversity** assesses how distinct the generated sequences are from each other. We run MMseqs2 with the generated set as both query and target, compute dissimilarity for all sequence pairs (excluding self-pairs), and define the diversity of each sequence as the minimum dissimilarity to any other sequence in the set. We use the same approach to evaluate diversity in protein optimization outputs.

**Novelty** measures how different the generated sequences are from a reference set. In our protein generation experiments, the reference set is all lysozyme-like proteins in the UniRef50 (Suzek et al., 2015) dataset. We use MMseqs2 to align generated sequences (query) to the reference set (target), and compute novelty as the average dissimilarity for each generated sequence against all reference sequences.

**Dissimilarity to Initial Set** (**Dissim$_{init}$**) quantifies how different the optimized sequences are from the initial set. We compute the average dissimilarity of each optimized sequence to all sequences in the initial set, following the same procedure as for novelty.

**Dissimilarity to High-Fitness Set** (**Dissim$_{high}$**) is defined similarly to **Dissim$_{init}$**, but uses the high-fitness set as the reference set.

## A.2. Fitness Predictor

**Thermostability Predictor.** We construct a thermostability predictor using ESM2-650M as the feature extractor. The predictor adopts the same architecture as the "lm_head" in ESM2-650M and is trained with mean squared error loss. For training, we use data from the Meltome Atlas (Jarzab et al., 2020), which provides melting temperatures for 48,000 proteins across 13 species (archaea to humans), with values ranging from 30°C to 90°C. To focus on sequence-dependent effects and minimize species-specific variation, we use the median melting temperature across all species for each protein as its final label.

The dataset is split into 90% for training and 10% for testing. To reduce redundancy, we ensure a maximum sequence identity of 90% within the training set. Furthermore, any training sequence with $\geq$30% identity to a test sequence is removed, preventing information leakage and ensuring a fair evaluation. The final dataset contains 24,817 proteins for training and 3,134 for testing.

On the test set, the predictor achieves a Spearman rank correlation of 0.76.

**Solubility Predictor.** The solubility predictor is a binary classifier with the same architecture and training procedure as the thermostability predictor. We use the preprocessed dataset in khurana2018deepsol, containing 28,972 soluble and 40,448 insoluble proteins. The data is split 90%/10% for training and validation. For benchmarking, we use an independent test set in (Chang et al., 2014), which includes 1,000 soluble and 1,001 insoluble proteins.

On this test set, our predictor achieves an accuracy of 0.708, precision of 0.758, recall of 0.612, and F1 score of 0.677, demonstrating its effectiveness for sequence-based solubility prediction.

*Table 5.* Comparison of generating lysozyme-like protein with both high thermostability and solubility. Results are reported as mean (std).

| | | **Thermostability** | **Solubility** | **Diversity** | **Novelty** |
|---|---|---|---|---|---|
| **ESM2** | Original Model | 56.45 (11.07) | 0.328 (0.151) | 0.967 (0.019) | 0.596 (0.136) |
| | Fine-tuning | 59.69 (13.22) | 0.406 (0.199) | 0.966 (0.020) | 0.596 (0.138) |
| | Activation Steering | **68.02** (14.44) | **0.483** (0.244) | **0.992** (0.005) | **0.950** (0.070) |
| **ESM3** | Original Model | 54.46 (10.48) | 0.314 (0.193) | 0.962 (0.016) | 0.572 (0.122) |
| | Fine-tuning | 60.22 (13.90) | 0.425 (0.206) | 0.960 (0.016) | 0.568 (0.122) |
| | Activation Steering | **66.75** (9.61) | **0.451** (0.253) | **0.980** (0.009) | **0.925** (0.108) |

*Table 6.* Comparison of generating lysozyme-like protein with high thermostability or solubility. Results are reported as mean (std).

| **Base Model** | **Method** | **Thermostability** | | | **Solubility** | | |
|---|---|---|---|---|---|---|---|
| | | **Thermostability ↑** | **Diversity ↑** | **Novelty ↑** | **Solubility ↑** | **Diversity ↑** | **Novelty ↑** |
| ESM2-650M | Original Model | 56.48 (12.04) | 0.954 (0.023) | 0.591 (0.110) | 0.289 (0.151) | 0.963 (0.019) | 0.596 (0.130) |
| | Fine-tuning | 63.56 (14.87) | 0.953 (0.023) | 0.585 (0.099) | 0.356 (0.213) | 0.961 (0.020) | 0.594 (0.132) |
| | Activation Steering | **82.20 (12.92)** | **0.971 (0.023)** | **0.739 (0.130)** | **0.494 (0.241)** | **0.998 (0.001)** | **0.880 (0.087)** |
| ESM2 3B | Original Model | 56.05 (11.24) | 0.968 (0.020) | 0.632 (0.143) | 0.298 (0.174) | 0.971 (0.021) | 0.622 (0.162) |
| | Fine-tuning | 64.22 (14.49) | 0.965 (0.022) | 0.629 (0.143) | 0.385 (0.236) | 0.966 (0.022) | 0.610 (0.165) |
| | Activation Steering | **83.33 (9.47)** | **0.990 (0.006)** | **0.915 (0.105)** | **0.631 (0.228)** | **0.996 (0.003)** | **0.951 (0.077)** |

# B. Additional Experiment Results

### B.1. Activation Steering for Multiple Desired Properties

Previous experiments demonstrated the effectiveness of Activation Steering for guiding protein generation toward a single desired property. In this part, we extend our evaluation to the simultaneous optimization of multiple properties. Specifically, we aim to lysozyme-like proteins with both high thermostability and solubility.

For our Activation Steering, we compute the steering vectors for thermostability and solubility as $v_l^{\text{therm}}$ and $v_l^{\text{sol}}$, respectively. We then obtain the steering vectors for performing activation addition in Equation (1) as $v_l = 0.5 v_l^{\text{therm}} + v_l^{\text{sol}}$. For the fine-tuning baseline, we fine-tune the model using the union of positive data sets for these two properties.

Table 5 shows that Activation Steering consistently outperforms both the original model and fine-tuning across all metrics and backbones (ESM2 and ESM3). For instance, on ESM2, Activation Steering improves thermostability from 56.45 to 68.02, and solubility from 0.328 (original) and 0.406 (fine-tuned) to 0.483. Similar trends hold for ESM3. Although gains are smaller when optimizing multiple properties at once, Activation Steering remains more effective than fine-tuning or the original model for jointly optimizing multiple protein properties, without reducing diversity or novelty.

### B.2. Activation Steering on Larger PLM

To assess the scalability of our activation steering method, we evaluate its performance on the larger ESM2-3B model, using ESM2-650M as a reference. The task settings and baselines are the same as the protein generation experiments described in Section 4.1.

Table 6 summarizes the results. Compared to the smaller ESM2-650M, ESM2-3B consistently achieves higher scores across all metrics, demonstrating the benefits of scaling up the model size for protein generation tasks.

On ESM2-3B, Activation Steering achieves the best performance for both thermostability and solubility, with mean values of 83.33 and 0.631, respectively. These improvements are accompanied by substantial gains in diversity and novelty. The performance gap between Activation Steering and the baselines is even more pronounced on ESM2-3B than on ESM2-650M, indicating that the effectiveness of Activation Steering is amplified as the model size increases. This suggests that large protein language models are better able to leverage activation-based steering for generating diverse and novel sequences with improved target properties.

In summary, these results demonstrate that Activation Steering remains effective as model size increases, and that scaling up the model further enhances its ability to generate proteins with desirable characteristics.

