# OpenReview forum: "Steering Protein Language Models"
_ICML.cc/2025/Conference — ICML 2025 poster_

### Official Review · Reviewer_Vbjw · 2025-02-23

**Overall Recommendation:** 3

**Summary:**

This paper presents a method to control the output of protein language models, inspired by the Activation Steering approach in the LLM domain, allowing the generated sequences to exhibit a given property. This method does not require retraining the protein language model and can be directly applied during the inference stage. The paper validates the effectiveness of this approach by manipulating properties such as thermostability and solubility.

**Claims And Evidence:**

The paper experimentally demonstrates that the proposed method can generate sequences with improved performance on a given property, supporting its claim.

**Essential References Not Discussed:**

I did not find any important missing references.

**Experimental Designs Or Analyses:**

I have reviewed the experimental design section of the paper, and I believe more testing metrics need to be included. The paper primarily tests how well the generated sequences from the protein language model align with the target properties and measures diversity and novelty, but I believe an important metric for evaluating PLM-generated results is testing the authenticity of these sequences, which the authors have overlooked. It is necessary for the paper to include metrics that assess the authenticity of the generated sequences, such as pLDDT and sequence likelihood.

**Methods And Evaluation Criteria:**

The method proposed in this paper is reasonable.

However, I believe the evaluation criteria used in this paper have potential issues. During the generation process of ESM-2, the model's behavior (such as activation steering or certain activation parameters) directly depends on the structure and inference process of ESM-2, and the generated protein is then evaluated for properties such as thermostability using ESM-2. If the activation of the model is guided or modified during the generation process, then the evaluation may exhibit a certain level of circular dependency with the generation process, potentially causing the generated protein to perform exceptionally well in the ESM-2 evaluation, because the model has already been adjusted during the generation. This could potentially be considered an attack on ESM-2. In this part of the experiment (Table 1), we also observe that ESM-2 shows the best performance. Therefore, the paper needs to clarify whether this good performance is due to such an attack.

**Other Comments Or Suggestions:**

I don't have other comments.

**Other Strengths And Weaknesses:**

I am curious whether the paper's method performs well for modeling more niche properties. The model's performance was primarily validated in terms of thermostability, solubility, and fluorescence brightness, but these are already well-studied properties, and many protein sequences with these properties are included in the PLM training data. However, for protein design tasks involving more niche properties, I am uncertain whether this method would still perform well, as it fundamentally relies on the internal knowledge of PLM.

**Questions For Authors:**

I don't have other questions.

**Relation To Broader Scientific Literature:**

The paper proposes a more novel method for controlling PLM output, which is insightful for protein design.

**Theoretical Claims:**

This paper is more "application-oriented," and therefore does not include many theoretical claims.

---

> ### Author Rebuttal · Authors · 2025-04-01
>
> We sincerely thank the reviewer for providing valuable feedback. We detail our response below point by point. Please kindly let us know whether you have any further concerns.
>
> ----
>
> **Q1**: "the evaluation may exhibit a certain level of **circular dependency**. ... This could potentially be considered an attack on ESM-2. In this part of the experiment (Table 1), we also observe that ESM-2 shows the best performance. "
>
> We thank the reviewer for raising the important issue of potential circular dependency in our evaluation. Using ESM-2 both as the base generative model and as part of the downstream predictor could potentially introduce biases. However, we emphasize the following points to address this concern:
>
> - While using ESM-2 as a base model achieves superior performance in Table 1 to other base models, by comparing the generation performance with Action Steering and without Activation Steering (Original Model), we can we can observe a significant performance gain. This validates that improvements are attributable to the steering mechanism rather than inherent biases in ESM-2.
>
> - Besides, our proposed Activation Steering method demonstrates consistent and significant improvements over both Fine-tuning and the Original Model across all three base models (ESM-2, ESM-3, and ProLLaMA). This cross-model consistency suggests the improvements stem from the method itself rather than inherent biases in ESM-2.
>
> -----
>
> **Q2**: " It is necessary for the paper to include metrics that assess the authenticity of the generated sequences, such as pLDDT and sequence likelihood."
>
> To address the concern about assessing the authenticity of generated sequences, we incorporate the pLDDT metric, evaluated using ESMFold, to measure the structural integrity of both initial and optimized protein sequences. The results, presented in the tables below, demonstrate that the pLDDTs remain consistent before and after applying activation steering. This consistency confirms that our method effectively guides protein generation towards desired functionalities while maintaining the authenticity of the sequences
>
> **Table 1**: pLDDT for protein generation.
>
> | Base Model | Method | Thermostability | Solubility |
> | -- | -- | -- | -- |
> | ProLLaMA | Original Model | 40.02 | 40.02 |
> | ProLLaMA | Fine-tuning | 37.79 | 37.68 |
> | ProLLaMA | Activation Steering | 39.51 | 39.52|
> | ESM2 | Original Model | 74.09 | 69.17 |
> | ESM2 | Fine-tuning | 72.62 | 68.16 |
> | ESM2 | Activation Steering | 73.02 | 68.36 |
> | ESM3 | Original Model | 76.14 | 73.54 |
> | ESM3 | Fine-tuning | 75.50 | 72.71 |
> | ESM3 | Activation Steering | 76.18 | 72.61 |
>
> **Table 2** pLDDT for protein optimization regarding thermostability.
>
> | | Medium difficulty | Hard difficulty |
> | -- | -- | -- |
> | Before Optimization | 79.75 | 79.99 |
> | AdaLead | 51.06 | 31.15 |
> | ESM2 + ASPO | 76.75 | 79.59 |
> | ESM3 + ASPO | 81.00 | 78.91 |
>
> **Table 3** pLDDT for protein optimization regarding solubility.
>
> | | Medium difficulty | Hard difficulty |
> | -- | -- | -- |
> | Before Optimization | 77.92 | 75.94 |
> | AdaLead | 36.87 | 35.89 |
> | ESM2 + ASPO | 78.38 | 77.13 |
> | ESM3 + ASPO | 78.15 | 77.98 |
>
> ----
>
> **Q3:** "I am curious whether the paper's method performs well for modeling more **niche properties**... as it fundamentally relies on the internal knowledge of PLM."
>
> - We appreciate the reviewer's interest in the applicability of our method to more niche properties. We would first note that the challenge with niche properties is the scarcity of labeled data, which can hinder the training of effective predictors to estimate the performance.
>
> - We conducted an additional experiment focusing on the niche property of hydrolysis activity at 30°C for polyethylene terephthalate (PET). We utilized the dataset from (Seo et al. 2025), comprising 184 annotated samples.
>     - To estimate the performance, we train a predictor on this dataset, achieving a Pearson correlation coefficient (R) of 0.57 in 5-fold cross-validation, indicating a reasonable prediction capability under data constraints.
>     - We design optimization tasks of varying difficulty:
>         - Medium Difficulty Task: Optimizing 69 samples with an initial average predicted activity of 1.61.
>         - Hard Difficulty Task: Optimizing 67 samples with an initial average predicted activity of 175.62.
> - The results post-optimization using our ASPO method combined with ESM3 are as follows:
>
> |Task Difficulty|Initial Average Activity|Post-Optimization Average Activity|
> |-|-|-|
> | Medium | 1.61 | 5.00 |
> | Hard | 175.62 | 236.91 |
>
> - These results demonstrate that our ASPO method can effectively optimize even niche protein properties, such as hydrolysis activity in PET, underlining the versatility and potential of our method in broader protein design applications.
>
> > Reference: Seo, Hogyun, et al. Landscape profiling of PET depolymerases using a natural sequence cluster framework. Science 2025.

---

> > ### Comment · Reviewer_Vbjw · 2025-04-02
> >
> > Thanks for your rebuttal and I have raised my score to 3.

---

### Official Review · Reviewer_ADRt · 2025-03-09

**Overall Recommendation:** 3

**Summary:**

This paper introduces a method for steering Protein Language Models (PLMs) towards outputs with desirable properties. It is based on a technique called 'Activation Steering' used for LLMs, where internal activation vectors of LLMs are modified to shift them towards desired behaviour.

The authors show how this method can be applied to both Autoencoder PLMs  like ESM2 and ESM3 as well as Autoregressive PLMs like ProLLAMA to produce proteins with higher ‘fitness’, i.e. higher values of desired properties like thermostability and solubility. In line with previous work in this field, oracle models are created by training CNN models based on relatively large datasets.

The proposed method shows performance improvements over existing methods, while maintaining diversity and novelty of generated proteins.

**Claims And Evidence:**

1.	The paper claims to enable property-specific protein generation without the requirement for retraining models, thus providing a scalable alternative to other resource intensive techniques.
2.	The paper introduces a new iterative optimization method which involves choosing which tokens to mutate based on an estimate of their individual fitness score.
3.	An empirical evaluation is performed on models with 3 different architectures to show the universal effectiveness of the method and its performance in comparison to some existing methods in the field.

Claim (1) is supported by empirical evidence, at least for the models included in the paper. Activation steering is indeed less resource intensive than directed evolution/reinforcement learning based methods as well as fine-tuning, though it is unclear if it so in comparison with other benchmarks used in the paper.

Some of the technical claims made in the paper are unconvincing to me, such as “magnitude of a representation’s projection onto [the direction of the steering vector] can indicate it’s relatedness to the property”, which I have detailed in the methods/questions section. The method itself also makes certain assumptions which I don’t think are true. Again, I have detailed these in later sections.

With respect to claim (2) – the paper does introduce a new optimization procedure to choose the tokens most weakly associated with the property of interest and update the internal representations of those tokens. It is unclear to me if the method of determining if a token contributes positively or negatively to the fitness function is completely correct, which follows from the point made above about projection onto the steering vector. Moreover, it is not clear to me from a theoretical standpoint how simply adding a common steering vector to the layer representation of any token, no matter where it is in relation to the regions of positive and negative fitness values, would improve overall fitness of the protein. Again, I have elaborated on this in the question section.
For claim (3), empirical evidence is shown on 3 different architectures. With the lack of any theoretical discussion or explanation I’m unconvinced on why this method universally applicable to all PLMs. Is there a fundamental property of models on which this method is based? Latent spaces of different models behave differently – some may be amenable to post-training modifications while some may not.

Overall, I think this paper may still be valuable since it details a method which empirically shows better results than existing methods for a real-world task with meaningful utility. However, there are multiple methodological leaps and assumptions which mean that it doesn’t convince me in terms of being an actual novel technical contribution.

**Essential References Not Discussed:**

NA

**Experimental Designs Or Analyses:**

Experiments are well designed to support the claims of the paper. Results are shown on major PLM structures (though only 3) and for three different properties.

**Methods And Evaluation Criteria:**

Methods:

I’m unsure about certain parts of the method, which I have detailed below:

1.	Firstly, the assumption that “PLMs inherently encapsulate intrinsic knowledge about specific protein properties” is only true if the training dataset has sufficiently large numbers of samples exhibiting the negative to positive range of values you want to generate at inference time. Without a meaningfully high number of samples, the model would not gain a semantic understanding of the required property and would not encode this in the structure of its latent space. This is not true for fine-tuning based approaches, where you can fine-tune on an additional dataset with relatively few samples and get desired behaviour. Thus, this method is an alternative to fine-tuning based methods only when the property of interest is already in the dataset the model is trained on.
2.	Secondly, I’m unsure whether just adding a vector to the latent representation of a deep learning model will always lead to meaningful results. For example, adding a random vector to the latent representation of the model is likely to lead to gibberish outputs, as demonstrated in many papers on adversarial attacks. What if adding the steering vector to the latent representation takes the model to an unseen region, i.e. a region which doesn’t exist in the training set?
3.	Even if you assume that the model is well behaved under the steering transformation, it is not strictly true that adding the steering vector to the latent representation of a particular sample will move it towards the desired property. Concepts like functional continuity, linearity, and other properties necessary for this to be true are not necessarily naturally developed by the model. As a simple thought experiment, assume that there is a spherical region in the middle of the latent space that corresponds to positive fitness. Then, in order to transform samples outside the positive space to lie within the positive space, I need to add vectors to all samples pointing inward towards the positive region, whereas the steering vector would always point in the same direction. You are assuming that a translational transform in the latent space of each layer can shift any token towards the desired property, which is not true.
4.	“the magnitude of a representation’s projection onto [the direction of the steering vector] can indicate its relatedness to the property”. This is not strictly true. For example, you can infinitely scale a vector and its projection onto the steering vector will keep increasing. That doesn’t necessarily mean it’s relatedness to the property will keep increasing.
5.	One question I have with this method is how you maintain existing properties of proteins not necessarily related to the fitness property, especially for protein optimization. This is especially possible with the iterative optimization where you repeatedly replace the tokens least associated with the property to optimize for. For example, say I have a protein with all the required properties but solubility. In optimizing for solubility using this method, do I lose other properties? This is important for protein optimization because I would generally start with a protein with certain properties and try to optimize for others. If the optimized protein is unrecognizable from the protein I start from, then this is just a conditional protein generation method, not an optimization method.

[Discuss all method drawbacks here]

Evaluation:
1.	In line with previous works, this paper uses a trained model as an oracle. This model is trained on thousands (~40000) labeled examples and shows medium to high accuracy/correlation on a test set. While this is not optimal, it is in line with previous work in the area published at similar venues. In the absence of actual ground-truth values for the propertied of interest, this evaluation criteria, though not perfect, is the best that is possible.
2. Apart from fitness, evaluation criteria, such as diversity and novelty are well-chosen to assess the effectiveness of the generated sequences. One thing that is missing is, as mentioned, maintenance of existing properties apart from the property to optimize, as mentioned previously.
2.	I’m unclear on the Dist_init and Dist_high metrics – are large or small values of these metrics favourable? I’m not sure how these metrics contribute to telling us about the quality of the output. There’s also no clear pattern in Tables 2, 3, 4 that tells me which models are better.

**Other Comments Or Suggestions:**

I will rate this paper as a 'weak accept' but I think certain issues need to be addressed, mainly point 2, 3 listed in the 'Methods and Evaluation' section above.

**Other Strengths And Weaknesses:**

Strengths:

1. The paper addresses a topic of importance in protein engineering.
2. The method is conceptually simple and seemingly shows good results despite its simplicity.
3. The method is demonstrated on both AE-PLMs and AR-PLMs, implying some sort of generalizability. However, this is not surprising given that the method is based on a fundamental concept of semantic organization in neural network latent spaces, which could be counted as a strength of the method.

Weaknesses:

I have outlines multiple weaknesses related to the method in the 'Methods and Evaluation' section. I will briefly summarize them here:

1. Optimizing for properties which are not reflected in the original training set. This is only a criticism because the method is compared to fine-tuning, which can be used to generate proteins with properties not in the original set using a small, new dataset.
2. Assumptions about the steering transformation being a coherent transformation path in the feature space of the model, in terms of semantic continuity of the space, linearity, etc. Theoretical grounding would be useful here.
3. Concerns about the magnitude of the projection of a vector reflecting it's relatedness to the property. Counterexample given in the 'Methods and Evaluation' section (point 4).
4. Questions about maintaining existing properties unrelated to the property to optimize for.
5. Model evaluation is based on oracle classification models without actual ground truth. However, as I mentioned, there's no real way to get around this short of actual experimental validation.

**Questions For Authors:**

Most of my questions are listed in the sections above.

**Relation To Broader Scientific Literature:**

The paper is in line with previous work on protein fitness optimization.

**Theoretical Claims:**

No major theoretical claims are made, the paper is mostly based on empirical results. It would be good to see some theory on major claims, such as whether adding steering vectors to intermediate layers always moves the output closer to the high fitness region, and whether this happens in a linear or smooth manner from the initial point to the final point as you perturb the activations more and more.

---

> ### Author Rebuttal · Authors · 2025-04-01
>
> Thank you for your thoughtful and positive feedback. We have provided a detailed explanation for your concerns as follows. Please feel free to let us know if you have any additional concerns or questions.
>
> **W2: Assumptions .. continuity, linearity...Theoretical grounding**
>
> - Our method is based on two hypotheses: the **linear representation hypothesis** and the **superposition hypothesis** (T. Adly 2024). The linear representation hypothesis suggests that neural networks encode meaningful concepts as directions in their activation spaces, while the superposition hypothesis extends this by suggesting that networks utilize almost-orthogonal directions in high-dimensional spaces to represent features, embodying properties of additivity and homogeneity. These hypotheses underpin the design of many existing algorithms.
> >Ref: T. Adly. Scaling monosemanticity: Extracting interpretable features from Claude 3 sonnet. Anthropic, 2024.
>
> - However, the theoretical validation of these hypotheses is ongoing. As such, the theoretical grounding of our method remains a subject for future research.
>
> **M1: the assumption .. only true if dataset has large numbers of samples**
>
> - Our method assumes that the pretrained PLM has already encapsulated comprehensive knowledge of protein properties, given its pretraining on a massive number of both naturally occurring and synthetically designed protein sequences. This extensive pretraining dataset supports our assumption that the PLM possesses a universal understanding of various protein properties, making it suitable for steering the PLM to explore specific properties.
>
> - However, we acknowledge that if the PLM lacks prior knowledge of a target property, our method may not be effective, which is a limitation.
>
> - Regarding the reviewer's point on fine-tuning, it is important to clarify that FT also relies on the pretrained model having some foundational knowledge of the desired properties. Without this, FT on a small dataset risks overfitting and poor generalization, similar to the limitations faced by our proposed method.
>
> **M2: unsure whether just adding a vector**
>
> - We appreciate the reviewer's concern regarding the potential risks of modifying latent representations. Indeed, adding arbitrary vectors can disrupt model outputs, akin to adversarial attacks. However, our method carefully defines the steering vector to ensure meaningful modifications aligned with desired properties.
>
> - To illustrate, consider a simple case with a positive sample ($z_p$) and a negative sample ($z_n$). To shift $z_n$ towards $z_p$, the intuitive direction for modification is $v=z_p-z_n$. Extending this, if we have sets of positive and negative samples, the steering vector can be defined as the mean difference between these sets, aligning changes with the observed data distribution.
>
> - Our method assumes that the latent space adheres to the linear representation hypothesis and superposition hypothesis, allowing these mean-based modifications to navigate towards desired properties effectively. Empirical results confirm that this method in steering the latent representation leads to the desired enhancements.
>
> **M3: not strictly true that adding the steering vector .. move it towards the desired property**
>
> - The effectiveness of the proposed method relies on the linear representation hypothesis and superposition hypothesis. The thought experiment designed by the reviewer, where the latent space exhibits non-linear characteristics, poses a challenge to our assumptions. In this case, the proposed method does not work.
>
> **M4&W3: Concerns about the magnitude .. reflecting relatedness to the property**
>
> - Normalization techniques like LayerNorm and RMSNorm in transformers constrain vector magnitudes, ensuring they don’t scale infinitely. This addresses the counterexample.
>
> - Additionally, the magnitude is indicative of the importance of the token. Our relatedness score takes the magnitude of the token representations into consideration, following the attention mechanisms in transformers, which compute attention scores via dot products between queries and tokens.
>
> **M5&W4: maintain unrelated properties**
>
> - We evaluated both thermostability and solubility to ensure our method maintains unrelated properties during optimization. Due to length limitations, we only present solubility experiments. As shown in the following tables, steering for solubility has minimal impact on the unrelated property thermostability.
>
> Tab: Protein generation
> |Base Model |Method|sol(target)|therm(unrelated)|
> |-|-|-|-|
> | ProLLaMA | Original|.23|56|
> |ProLLaMA|FT|.24|57|
> |ProLLaMA|AS|.28|56|
> |ESM2|Original|.33|57|
> |ESM2|FT|.41|56|
> |ESM2|AS|.44|57|
> |ESM3|Original|.32|54|
> |ESM3|FT|.39|55|
> |ESM3|AS|.49|54|
>
> Tab: Protein optimization
> ||Medium|Medium|Hard|Hard|
> |-|-|-|-|-|
> ||**sol**(target)|**therm**(unrelated)|**sol**|**therm**|
> | Before Opt|.28|54|.09|54|
> | AdaLead|.62|50|.53|51|
> | ESM2+ASPO|.51|53|.35|54|
> | ESM3+ASPO|.65|53|.4|53|

---

### Official Review · Reviewer_fV4m · 2025-03-14

**Overall Recommendation:** 3

**Summary:**

This paper adapts activation steering techniques from LLMs to Protein Language Models (PLMs) to guide protein generation toward desired properties without retraining. It introduces an Activation Steering-based Protein Optimization (ASPO) framework that outperforms existing methods on thermostability, solubility, and GFP brightness tasks.

**Claims And Evidence:**

The key claims are well-supported by experimental evidence across multiple PLM architectures. Results show significant improvements in target properties while maintaining diversity and novelty compared to fine-tuning and other baselines.

However, the worse fine-tuning results might be due to insufficient data and large numbers of learnable parameters. Do you use parameter-efficient fine-tuning, like LoRA?

**Essential References Not Discussed:**

N/A

**Experimental Designs Or Analyses:**

The number of use cases is small though.

In the ablation studies, it's better to show the trend of both properties and basic protein generation qualities with respect to the ablated variables.

**Methods And Evaluation Criteria:**

The methodology is sound, with appropriate evaluation metrics (fitness, diversity, novelty, distance measures) and comprehensive comparison against established baselines on relevant protein engineering tasks.

**Other Comments Or Suggestions:**

See above.

**Other Strengths And Weaknesses:**

See above.

**Questions For Authors:**

The single objective design is quite limited. How do you deal with multi-property optimization?

**Relation To Broader Scientific Literature:**

It should be easy for the people in science community to adapt this method with a pre-trained large models.

**Theoretical Claims:**

N/A

---

> ### Author Rebuttal · Authors · 2025-04-01
>
> We appreciate very much your constructive comments on our paper. We have provided a detailed explanation for your questions as follows. Please feel free to let us know if you have any additional concerns or questions.
>
> ----
>
> **Q1: "How do you deal with multi-property optimization?"**
>
> Thank you for your insightful question regarding our method for multi-property optimization..
>
> - We propose a simple solution to perform steering on multiple properties by computing a composite steering vector that incorporates the steering vectors of individual properties. Specifically, if $v_\ell^{therm}$  represents the steering vector for thermostability
> and  $v^{sol}_\ell$ for solubility, the combined steering vector used for multi-property optimization is given by
>
> $v_\ell = v_\ell^{therm} + v^{sol}_{\ell}$.
>
> - The remaining settings are the same as single property optimization.
> - We empirically test this method in scenarios involving both protein generation and protein optimization. The results, as detailed in the tables below, demonstrate that our method effectively improves multiple properties, albeit with a slight trade-off compared to optimizing a single property.
> - We plan to further refine our method to better manage these trade-offs and explore more sophisticated methods for steering vector combination and weighting for multi-property optimization in future work.
>
> **Table 1**: Multi-Property Steering for Protein Generation
>
> | Based Model|Method|Thermostability|Solubility |
> | --|--|--|-- |
> | ESM2|Original Model|56.77|0.327 |
> | ESM2|Activation Steering|75.29|0.409 |
> | ESM3|Original Model|60.70|0.312 |
> | ESM3|Activation Steering|67.82|0.449 |
>
>
> **Table 2**: Multi-Property Steering for Protein Optimization on Thermostability Medium Difficulty Task
>
> | Method|Thermostability|Solubility |
> | --|--|-- |
> | Before Optimization |59.78|0.299 |
> | ESM2 + ASPO|76.19|0.401 |
> | ESM3 + ASPO|76.64|0.332 |
>
> **Table 3**: Multi-Property Steering for Protein Optimization on Thermostability Hard Difficulty Task
> | Method|Thermostability|Solubility |
> | --|--|-- |
> | Before Optimization |46.38|0.237 |
> | ESM2 + ASPO|78.11|0.278 |
> | ESM3 + ASPO|74.99|0.311 |
>
> ----
>
> **Q2: "In the ablation studies, it's better to show the trend of both properties and basic protein generation qualities with respect to the ablated variables."**
>
> Thank you for your valuable suggestion. We include trends for diversity, distance to the initial set, and distance tothe  high fitness set in the following tables.  and will update them in the corresponding figures. We will also reflect these trends in the revised figures to provide a clearer visualization of how the ablated variables impact both properties and basic protein generation qualities.
>
> **Table 4**: Sensitivity to number of samples for steering vectors extraction in protein thermostability optimization (Fig. 4(a))
>
> | number of samples|10|25| 50|100|250 |
> | --|--|--|--|-- |-- |
> | ESM2-Medium-Diverisity|6.21|7.69|7.86|7.71 |7.71 |
> | ESM2-Medium-Dist_init |6.15|6.75|7.49|7.709|8.15 |
> | ESM2-Medium-Dist_high |10.87|10.77|10.54|10.63|10.51 |
> | ESM2-Hard-Diverisity|4.82|6.04|5.97|6.09|6.13 |
> | ESM2-Hard-Dist_init |6.66|7.21|7.02|7.29|7.30 |
> | ESM2-Hard-Dist_high |10.21|10.44|9.91|8.825 |9.10 |
> | ESM3-Medium-Diverisity|6.96|6.96|6.92|6.94|6.94 |
> | ESM3-Medium-Dist_init |6.08|6.09|6.09|6.00|6.00 |
> | ESM3-Medium-Dist_high |10.53|10.31|10.08|9.71 |10.12 |
> | ESM3-Hard-Diverisity|6.93|6.95|6.91|6.92 |6.932 |
> | ESM3-Hard-Dist_init |7.71|7.60|7.60|7.57|7.68 |
> | ESM3-Hard-Dist_high |9.94|9.74|9.46|9.25|9.27 |
>
>
> **Table 5**: Sensitivity of $\alpha$ in protein thermostability optimization (Fig. 6(a))
>
> | $\alpha$|0.05|0.25|0.5|1|2|5|20 |
> | --|--|--|--|--|--|--|--|
> | ESM2-Medium-Diverisity|7.01|7.68 |7.70|7.71|7.29|7.79|7.74 |
> | ESM2-Medium-Dist_init |7.67|7.68|7.71|7.71|8.00|7.88|7.89 |
> | ESM2-Medium-Dist_high |10.43|10.45|10.52|10.63|11.42|11.46|11.39 |
> | ESM2-Hard-Diverisity|6.83|6.05|6.07|6.09|6.348|6.370|6.348 |
> | ESM2-Hard-Dist_init |6.43|7.68|7.71|7.29|7.37|7.36|7.36 |
> | ESM2-Hard-Dist_high |8.88|8.88|8.86|8.83|9.19|9.15|9.19 |
> | ESM3-Medium-Diverisity|7.01|7.25|7.14|6.94 |6.91|6.91 |6.91 |
> | ESM3-Medium-Dist_init |6.00|6.00|6.00|6.00|6.01|6.01|6.01 |
> | ESM3-Medium-Dist_high |9.66|9.63|9.66|9.71|10.06|10.06|10.06 |
> | ESM3-Hard-Diverisity|7.15|7.14|7.02|6.92 |6.83|6.83|6.83 |
> | ESM3-Hard-Dist_init |7.59|7.58|7.58|7.57|7.57|7.57|7.57 |
> | ESM3-Hard-Dist_high |9.38|9.32|9.28|9.25|9.84|9.84|9.84 |
>
> ----
>
> **Q3: "the worse fine-tuning results might be due to insufficient data and large numbers of learnable parameters. Do you use parameter-efficient fine-tuning, like LoRA?"**
>
> - We appreciate your comment regarding parameter efficiency in fine-tuning. Indeed, we employ LoRA, specifically with a rank of 8, which we found optimal in our experiments after evaluating various ranks [2, 4, 8, 12, 16]. The rank of 8 outperformed others, with rank 4 being the next best. Our choice of hyperparameter alpha is 16.

---

### Official Review · Reviewer_ApT8 · 2025-03-18

**Overall Recommendation:** 3

**Summary:**

This paper introduces activation steering, which is a technique adapted from large language models, to control protein language models for generating and optimizing protein sequences with targeted properties (e.g., thermostability, solubility, fluorescence).
The method modifies internal model activations using steering vectors derived from contrastive representations of proteins with desired and undesired properties. For optimization tasks, the authors propose ASPO (activation steering-based protein optimization) and identify critical mutation sites via projection onto steering vectors. Experiments across autoencoder (i.e., ESM2, ESM3) and autoregressive (i.e., ProLLaMA) protein LMs demonstrate significant improvements in target properties without model retraining, outperforming fine-tuning and traditional optimization baselines.

**Claims And Evidence:**

Mostly supported.

**Essential References Not Discussed:**

[PPLM] Plug and Play Language Models: A Simple Approach to Controlled Text Generation. In ICLR 2020.

**Experimental Designs Or Analyses:**

Yes.

**Methods And Evaluation Criteria:**

Yes.

**Other Comments Or Suggestions:**

**Minor:**

Typo: line 361, "Sensitivityive".

**Other Strengths And Weaknesses:**

**Strengths**

1. The proposed activation steering is a training-free powerful method to generate protein sequences with target properties.
2. The authors also propose a novel Activation Steering-based Protein Optimization (ASPO) framework to improve the protein optimization performance and identify the mutation site.
3. Experimental results demonstrate that the proposed approach is able to significantly improve the steering generation performance on target property across various protein language models (ESM2, ESM3, ProLLaMA).

**Weaknesses**

1. In section 4.1, the ESM3 model is able to generate a full sequence from all-mask states, while ESM2 can not. Therefore, for the ESM3 model, authors should also provide the results of directly generating full sequences in addition to revising based on a reference sequence.
2. Authors should provide more details about experiments, such as the number of iterations T, length distribution of generated sequences.
3. Clarity issue. Could you please further explain the lines 189-194: "For practical implementation, [...], A linear classifier to distinguish the representations from the desired and undersired sets."? Does this mean that you only steer the activation of the layer with the highest validation accuracy, instead of all layers, during inference?
4. Missing discussion of PPLM, an important related work on steering pre-trained LMs for conditional generation.

**Questions For Authors:**

1. ESM2/3 models both demonstrate the scalability that a larger scale model is able to obtain better performance in various tasks in their paper. Therefore, could the performance of steering be further boosted by enlarging the model scale (such as, using ESM2 3B/15B or ESM3 7B/98B)?
2. In Figure 3, for AR-PLM, as the number of samples for extracting steering vectors increases, the relevant property values consistently decline. The line graph in the figure starts at 10 samples, and I am curious about the performance when the number of samples is further reduced (e.g., 1 sample). If the final trend shows that performance is always decreasing as the number of samples increases, does this mean that for AR-PLMs, users should use as few samples as possible for steering to obtain the best result?
3. Considering that the proposed method is training-free, will the proposed steering approach be compatible with other protein steering or optimization methods?

**Relation To Broader Scientific Literature:**

n/a

**Theoretical Claims:**

Not applicable since there's no theoretical contributions.

---

> ### Author Rebuttal · Authors · 2025-04-01
>
> We sincerely thank the reviewer for providing valuable feedback. We detail our response below point by point.
>
> **W1: the ESM3 model ... generate a full sequence from all-mask states**
>
> We conduct experiments on full sequence generation using ESM3, maintaining the same settings as described in Section 4.1. The results are presented in the table below, demonstrating that Activation Steering significantly outperforms the baseline methods, thereby confirming its effectiveness.
>
> ||Thermostability|Solubility|
> |-|-|-|
> |Original Model|52.8|0.376|
> |Fine-tuning|65.5|0.412|
> |Activation Steering|79.6|0.466|
>
> **W2: more details about experiments**
>
> - The number of iterations T and the value of K
>     - Protein optimization: thermostability: T=8, K=4; solubility: T=4, K=2; GFP: T=4, K=2
> - Length distribution of generated sequences
>     - Protein generation
>         - thermostability: 60~256, mean: 194.1
>         - solubility: 47~256, mean: 168.5
>     - Protein optimization
>         - thermostability
>             - medium difficulty: 60~256, mean: 183.9
>             - hard: 102-256, mean: 208.2
>         - solubility
>             - medium: 71~256, mean: 180.4
>             - hard: 47~253, mean: 158.9
>         - GFP: length of all sequences is 237
>
> **W3: further explain the lines 189-194**
>
> Our method first involves computing a relatedness score for each token to identify which should be mutated, as discussed prior to the mentioned lines. In lines 189-194, we propose to determine the layer for computing the relatedness score as the one with the highest validation accuracy. This ensures that the most informative layer are utilized for token selection.
>
> After determining the tokens for mutation, we replace these tokens with a mask token. Importantly, activation steering is applied during inference across all layers, except the input layer, to steer the model's prediction on the masked tokens towards the desired property.
>
> We will revise lines 189-194 to enhance clarity on these points.
>
> **W4: Missing discussion of PPLM**
>
> Thank you for pointing out the omission of PPLM. PPLM indeed pioneers the concept of steering by modifying key-value pairs in the model's attention mechanism, guided by gradients from an attribute model. In contrast, our method employs a simpler activation steering approach that directly manipulates activations and does not require training an additional attribute model or updating steering vectors.
>
> We will include discussion of PPLM in the related work to highlight these distinctions.
>
> **Q1: Could the performance of steering be further boosted by enlarging the model scale**
>
> - We conducted experiments using ESM2-3B for protein sequence generation, maintaining the same settings as in Sec 4.1. The results are summarized in the table below.
>     - Compared to ESM2-650M, the proposed method Activation Steering shows similar performance in thermostability and significantly improves solubility.
>     - In contrast, fine-tuning performance decreases with larger ESM2, likely due to the need for more data to achieve optimal results.
>
> ||Thermostability|Solubility|
> |-|-|-|
> |Original Model|56.1|0.298|
> |FT|64.2|0.385|
> |AS|80.5|0.631|
>
> **Q2: the performance when the number of samples is further reduced**
>
> - We appreciate the reviewer's interest in the performance of AR-PLM with fewer samples. To address this, we conducted additional experiments using ProLLaMA with sample sizes ranging from 1 to 10. Each configuration was tested 10 times to mitigate randomness. The results, summarized in the table below, indicate that the optimal number of samples varies depending on the property being optimized. For thermostability, the best performance occurs with 8 samples, while for solubility, it peaks at 3 samples. Notably, the performance does not consistently improve with fewer samples; the lowest number of samples (1 sample) did not yield the best results.
> - This suggests that while reducing the number of samples can sometimes enhance performance, likely by focusing the generation on a narrower subcluster of proteins with desired properties, there is a trade-off in terms of robustness. Performance becomes less predictable and can vary significantly depending on the specific samples used to compute steering vectors.
> - In conclusion, while fewer samples can sometimes be beneficial, the optimal number of samples depends on the specific application and desired property, balancing performance with robustness.
>
> ||1|2|3|5|8|10|
> |-|-|-|-|-|-|-|
> |Thermostability|64.3|61.8|57.4|71.8|74.5|73.5|
> |Solubility|0.344|0.491|0.507|0.492|0.446|0.302|
>
> **Q3: compatible with other protein steering or optimization methods?**
>
> In this paper, we focus on studying steering for PLM and we apply PLM steering exclusively within the proposed ASPO method for protein optimization, positioning ASPO as a competitor rather than complementary to existing methods. Integration with other protein optimization strategies remains an interesting direction for future research.

---

> > ### Comment · Reviewer_ApT8 · 2025-04-03
> >
> > I really appreciate authors' efforts in addressing my concerns. I accordingly raise my score to 3. Please do incorporate all the discussion above in the final version.

---

### Decision · Program_Chairs · 2025-05-01

**Decision:**

Accept (poster)

**Comment:**

The paper adapts activation steering from LLMs to PLMs for protein sequence generation and optimization, showcasing a training-free method and the ASPO framework with experimental improvements over baselines. Reviewers initially had concerns like lack of certain experiment details, unclear method descriptions, theoretical assumptions, and issues with evaluation criteria. The authors responded with additional experiments (e.g., full sequence generation for ESM3, niche property experiments), clarifications (such as method steps and related work like PPLM), and evidence to address these concerns, like using LoRA for fine-tuning, showing multi-property optimization solutions, and mitigating risks of latent representation modifications. Reviewers reassessed positively, with some raising their scores. Considering the research's significance, novelty of the approach, and the authors' effective responses to feedback, the paper is recommended for acceptance, provided the authors incorporate all relevant discussions and clarifications in the final version.